# The Effects of Nutritional Interventions on the Cognitive Development of Preschool-Age Children: A Systematic Review

**DOI:** 10.3390/nu14030532

**Published:** 2022-01-26

**Authors:** Marina Roberts, Terezie Tolar-Peterson, Abby Reynolds, Caitlin Wall, Nicole Reeder, Gina Rico Mendez

**Affiliations:** 1Department of Food Science, Nutrition and Health Promotion, Mississippi State University, Starkville, MS 39762, USA; mr2447@msstate.edu (M.R.); amr503@msstate.edu (A.R.); cw2693@msstate.edu (C.W.); nr657@msstate.edu (N.R.); 2Social Science Research Center, Mississippi State University, Starkville, MS 39762, USA; gina.mendez@ssrc.msstate.edu

**Keywords:** child development, cognition, preschool, nutrition

## Abstract

The developing human brain requires all essential nutrients to form and to maintain its structure. Infant and child cognitive development is dependent on adequate nutrition. Children who do not receive sufficient nutrition are at high risk of exhibiting impaired cognitive skills. This systematic review aimed to examine the effects of nutritional interventions on cognitive outcomes of preschool-age children. PubMed, PsycInfo, Academic Search Complete, and Cochrane Library electronic databases were searched to identify Randomized Controlled Trials (RCTs) published after the year 2000. Studies assessing the effects of food-based, single, and multiple micronutrient interventions on the cognition of nourished and undernourished children aged 2–6 years were deemed eligible. A total of 12 trials were identified. Eight out of the twelve studies found significant positive effects on cognitive outcomes. Iron and multiple-micronutrients supplementation yield improvements in the cognitive abilities of undernourished preschool-age children. Increased fish consumption was found to have a beneficial effect in the cognitive outcomes of nourished children. On the other hand, B-vitamin, iodized salt, and guava powder interventions failed to display significant results. Findings of this review highlight the importance of adequate nutrition during preschool years, and the crucial role sufficient nutrition plays in cognitive development.

## 1. Introduction

### 1.1. Nutrients and Cognitive Development

Malnutrition is characterized by an imbalance between a person’s nutrient requirements and their nutrient consumption, and includes conditions of overnutrition and undernutrition [1,2,3]. Undernutrition is caused by an inadequate intake of energy, protein, or vitamins and minerals [2], and is a present-day global problem hindering the development of young children [4,5,6,7,8,9]. For young children, undernutrition can cause emaciation, stunting, and wasting, or various micronutrient deficiencies [1,2,3,4]. Worldwide, 149 million children are stunted and 45 million are wasted [4]. Inadequate protein and energy intake in childhood is directly associated with reduced growth, and is indicative of several psychosocial problems later in life [3,10]. Undernourished children also exhibit impaired development and decreased functional capacity [10]. Pediatric undernutrition is characterized by a lack of adequate weight gain, low weight per height, or low weight per length, and is a direct contributor to impaired cognitive skills [11,12,13].

The human brain requires all essential nutrients, including protein, fats, carbohydrates, vitamins, minerals, and water, to form and maintain its structure. Therefore, adequate nutrition is essential for brain development and function [14,15,16]. However, micronutrients, such as iron, zinc, choline, iodine, folate, B12, and long-chain polyunsaturated fatty acids (LC-PUFAs) have been identified to be particularly relevant to cognitive development [14]. Iron is essential for the development of neurological pathways in the brain that influence brain function [15,17,18,19]. During the first two years of life, children experience rapid growth, which increases their iron requirement and places them at a higher risk for iron deficiency anemia [20]. Iron deficiency or iron deficiency anemia can negatively impact overall intelligence and cognitive development, especially if it occurs in early childhood [15,19,21]. Zinc is an essential trace mineral present in the brain that contributes to cerebral structure and function [22]. Zinc deficiency during infancy is associated with motor development delays [23] and detrimental effects on attention and short-term memory [15]. Long-term zinc deficiency is associated with stunting [24,25]. Choline is essential for the structural integrity of cell membranes and myelination [9,26,27]. Animal studies have shown choline deficiency to adversely impact memory [15,28,29]. Yet, the effects of choline on cognition in humans are still not fully understood. Iodine is an essential mineral for thyroid hormone synthesis and is required for brain development [15]. Iodine deficiency can have detrimental effects on cognitive function, and is the primary cause of intellectual disability around the world [15,30]. Folate is a water-soluble vitamin needed for DNA and RNA synthesis and the formation of the nervous system [15,31,32]. Maternal folate deficiency during the early stages of pregnancy is associated with an increased incidence of congenital malformations, including spina bifida and anencephaly [15,33]. Vitamin B12 is a cofactor in numerous catalytic reactions required for neurotransmitter synthesis and functioning [31,34]. Studies have linked B12 deficiency to cerebral atrophy and neurological disorders [34,35]. Vitamin A plays a critical-essential role in visual function [14,15]. LC-PUFAs, specifically docosahexaenoic acid (DHA) and eicosapentaenoic acid (EPA), are required for brain growth and development [15]. Inadequate intake of LC-PFAs is associated with impaired neurodevelopment, visual recognition, and memory [36,37,38].

### 1.2. Nutritional Interventions during the Preschool Years and Cognitive Outcomes

The first 1000 days of life are a crucial brain development period in which adequate nutrition is vital for optimal growth and cognitive development [39,40,41]. This has been identified as a sensitive time in which children are most vulnerable to behavioral and cognitive deficits [39]. A systematic review focusing on the first 1000 days of life identified the important role macronutrients, such as protein and LC-PUFAs, play in optimizing brain development [7]. Specifically, protein-energy malnutrition in early life can impede adequate brain growth, resulting in smaller brains [7]. Another review showed maintenance of adequate iron and zinc status contributes to adequate growth in early life, as significant positive effects were seen on child weight-for-age *z*-score (WAZ) and weight-for-height *z*-score [42]. Furthermore, maternal or child supplementation with choline has also been shown to support normal brain development [43]. Since the identification of the first 1000 days of life as a crucial cognitive development period, policy makers have placed strong emphasis on implementing nutritional policies that promote the healthy brain development of infants and toddlers [7]. However, public policy often does not extend to preschool-age children, even though the second 1000 days of life also represent a critical time in children’s cognitive and behavioral growth [44]. Children experience the most dynamic developmental changes during the preschool years, and acquire important skills that contribute to school readiness [44]. In particular, working memory and attention control undergo rapid progress, having an extensive impact on children’s academic achievement in later years [44,45]. Cognitive development reached in preschool years often predicts later achievements in life [45,46,47]. However, few articles have explored the effects of nutritional interventions on the cognitive outcomes of preschool-aged children. Children who do not receive adequate nutrition and psychosocial stimulation are likely to underperform in school and to have poor levels of cognition and education, which are linked to low-income earnings later in life [46,48,49]. This systematic review aimed to synthesize and evaluate the impact of nutritional interventions on the cognitive outcomes of preschool-aged children. The effects of food-based, single, and multiple micronutrient supplementation interventions were considered, in order to explore the correlation between nutrition interventions and cognitive performance.

## 2. Methods

This systematic search of scientific literature was conducted in May 2021, following the Preferred Reporting Items for Systematic Reviews and Meta-Analyses (PRISMA) guidelines [50]. The search strategy was developed in collaboration with an agriculture and forest resources librarian and with the guidance of the Population, Intervention, Comparators, Outcome (PICO) framework [51,52] (Table A1). Figure 1 illustrates the PRISMA flow diagram portraying the stream of evidence in different phases of this review.

### 2.1. Search Strategy

The authors used PubMed, PsycInfo, Academic Search Complete, and Cochrane Library electronic databases. RCTs performed between 2000 and 2021 were retrieved. The following search terms were used: (nutrient* OR nutrition* OR micronutrient* OR macronutrient* OR diet* OR “meal diversity” OR “food intake”) AND (“child development” OR cognition OR focus OR brain OR attentiveness OR attention OR memory OR verbal OR vocabulary OR learning OR literacy OR neuro* OR problem-solving OR reasoning OR “school performance” OR school) AND (child* OR preschool* OR “school age” OR school-age). Search terms were adjusted for each database to optimize the record retrieval process (Table A2) with the guidance of the Cochrane highly sensitive search strategies for identifying randomized trials [53].

### 2.2. Data Extraction

One author worked autonomously to determine the eligibility of titles and abstracts retrieved by the initial computerized search. Two other authors worked independently to evaluate qualified full-text records. Eligibility conflicts between reviewers were solved through discussions and with the assistance of a fourth reviewer. A backward search was conducted, in which reference pages of eligible articles were reviewed to ensure no pertinent studies were ignored in this review. Inclusion criteria consisted of: RCTs conducted after the year 2000, and that assessed cognitive outcomes of subjects aged 2–6 years consequent to food fortification, supplementation, or food-based interventions. Trials featuring secondary interventions including psychosocial stimulation and anthelmintic treatment combined with nutritional interventions were deemed eligible if cognitive function was a designated primary outcome. No language, geographical, or study duration restrictions were imposed. Exclusion criteria comprised: cross-sectional studies, non-randomized controlled trials, small sample size RCTs (<60 subjects), trials conducted in disease-specific population, studies focusing on children older than 6 years or younger than 2 years of age, trials providing dietary interventions targeted to the first 1000 days of life, and those that only provided parent nutrition education as the nutritional intervention. Table 1 includes a detailed description of the inclusion and exclusion criteria.

### 2.3. Risk of Bias

Retrieved studies were assessed by two reviewers independently using the Quality Criteria Checklist (QCC) for Primary Research from the American Dietetic Association [54]. The risk of bias tool includes four questions on relevance and ten questions on validity to appraise the appropriateness of study designs and the quality of how the studies were conducted [54]. The items assessed by the QCC include the research question, subject selection, comparability of groups, withdrawals, blinding, intervention/exposure, outcomes, analysis, conclusion support, and the likelihood of bias. Each item was classified as “Yes”, “No”, or “Unclear.” Studies were classified as negative (−) if six or more validity questions were answered as “No.” A positive (+) sign was assigned if the first four items and most answers were “Yes.” If the answer to any of the first four items was “No” and other items indicated strengths, the study was classified as neutral (Ø).

## 3. Results

### 3.1. Selection of Studies

The comprehensive database search resulted in 14,453 records once duplicates were removed. After reviewing titles and abstracts, 13,239 irrelevant studies were removed. A total of 69 full-text articles were further assessed. This systematic review identified 12 RCTs that met the inclusion criteria, of which three articles were identified through a backward search of the reference list of included publications. All selected studies were published in English.

Due to the high heterogeneity among studies, a meta-analysis was not deemed appropriate for this review. Specifically, variations in the way cognitive outcomes were defined and measured, as well as differences in the type and length of nutritional interventions, created major interpretative challenges.

### 3.2. Description of Studies

Table 2 summarizes the characteristics of RCTs contained in this review. In summary, 50% of trials were conducted in developed countries [55,56,57,58,59,60] and 50% in low and middle-income countries (LMICs) [61,62,63,64,65,66]. Four experimental studies were conducted in rural areas with low socioeconomic status [61,62,64,66], and six studies in urban areas with high socioeconomic status [55,56,57,58,59,60]. The earliest trial was completed in 2004 [58] and the latest in 2020 [65]. The shortest intervention was implemented for two months [58], and the longest trial lasted ten months [61].

### 3.3. Study Quality

Six studies had an overall low risk of bias [55,56,59,60,65,66] and six had a moderate risk of bias [57,58,61,62,63,64] as indicated in Table 3. Five studies did not record methods of handling withdrawal [57,58,61,62,63] and two studies [57,64] reported moderate differences between study groups at baseline.

### 3.4. Study Participants

The study populations were comprised mostly of preschool-age healthy children. However, children with insufficient folate levels, children at risk of undernutrition and micronutrient deficiencies, children with anemia, and children receiving anthelmintic medication were also included.

### 3.5. Nutritional Interventions

Supplement-based interventions were adopted in five out of twelve studies. These interventions included guava supplementation [62], DHA tablets [59], iron supplementation [58], B vitamin sachets [55], and iodized salt [61]. Multiple-micronutrient (MMN) food fortification interventions were implemented in three studies. For MMN interventions, the fortification was added to maize-porridge [64], rice, or wheat [65], and provided as a raw paste [66]. A total of three studies included food-based nutritional interventions in which subjects consumed fatty fish with daily meals [56,57,60]. Finally, one study conducted a dietary intervention in the form of a fortified milk powder [63], combined with cognitive stimulation.

### 3.6. Cognitive Tests

A diverse range of standardized cognitive tests were used to assess outcomes in multiple cognitive domains, including learning abilities, verbal reasoning, intellectual functioning, information processing speed, vocabulary, word reasoning, speed and accuracy of discrimination, fine and gross motor skills, coding, symbol search, and working memory. Half of the studies administered the Wechsler Pre-school and Primary Scale of Intelligence (WPPSI) test [55,56,57,60,61,63], two experimental studies administered the 9-Hole Peg Test (9-HPT) [56,57], and two studies administered the Mullen Scale of Early Learning (MSEL) [59,62]. Table 2 includes a detailed description of cognitive tests used in each trial, cognitive domains assessed, and effects on cognitive outcomes of preschool-aged children.

### 3.7. Major Cognitive Outcomes

#### 3.7.1. Single Nutrient Supplementation

Five RCTs measured the effect of supplement-based interventions on children’s cognition. Three trials failed to find a significant impact on cognition: the B vitamin [55], iodized salt [59], and guava [62] supplementation interventions. Guava is a fruit high in several vitamins and minerals and is also rich in lycopene, a carotenoid phytonutrient known for its antioxidant effect [67]. Guava was used as an intervention due to its high vitamin C content and its effects in facilitating nonheme iron absorption [68]. Although Guava supplementation yielded significant improvements in the iron status of children, no significant effects were seen in cognitive function [68]. Out of the two trials that found significant results, one of them was the iron intervention [58], which found that for children with iron deficiency anemia, iron supplementation increased accuracy and the speed of discrimination on the continuous performance task; however, for children with adequate iron status at baseline, iron supplementation did not affect performance on the continuous processing task. The DHA supplementation intervention [59] did not find any significant differences in cognitive function scores between the intervention and placebo groups; however, higher blood DHA levels were significantly associated with higher scores on the Peabody Picture Vocabulary Test, which measures listening comprehension and vocabulary.

#### 3.7.2. Multiple-Micronutrient Supplementation

Three RCTs measured the impact of multiple-micronutrient food fortification on children’s cognitive development. In the 11-week intervention consisting of eight grams of a point of use multiple micronutrient powder added to maize-meal porridge at breakfast for children 36–79 months of age [64], children in the intervention group significantly increased their scores on the simultaneous scale and the non-verbal index of the Kaufman Assessment Battery for Children compared to the control group. Conversely, in another longer trial where the children were somewhat younger [65], the cognitive benefits of a point of use multiple micronutrient powder were dependent upon whether the child was attending a low-quality or a high-quality preschool. Here, the quality of preschools was assessed by combining two validated scales, the Early Childhood Environment Rating Scale-Revised and the Home Observation for the Measurement of the Environment (HOME) adjusted for teachers rather than parents. Preschool quality was determined by two psychologists who independently assessed preschools as low or high-quality based on playing space, learning opportunities available to children, organization of environment, teacher-child interaction levels, and involvement levels of preschool-age children to activities and practices offered. Children from low-quality preschools who received the micronutrient powder fortification displayed improvements in expressive language and some, but to a lesser degree, improvement in inhibitory control and social-emotional development, whereas children attending high-quality pre-schools showed no improvements in cognitive outcomes. In the New Multicomponent Supplementary Food (NEWSUP) study [66], children 15–48 months of age were given a unique supplementary food that contained not just multiple micronutrients, but also plant polyphenols, omega-3 fatty acids, and protein for 23 weeks. Among the children younger than four years of age, the intervention group significantly increased working memory compared to the control group. However, among children 4–7 years of age who were administered the same intervention, no improvements in working memory were seen.

#### 3.7.3. Food-Based Interventions

Three RCTs evaluated the impact of food-based interventions on children’s cognition. All trials involved the provision of meals containing fatty fish. Among children aged 4-6 years enrolled in kindergartens in Germany, children who consumed Atlantic salmon three times per week for 16 weeks saw modest improvements in two indicators of non-verbal fluid intelligence that were greater than the improvements seen in the control group who received beef in place of salmon [56]. The salmon intervention did not improve total IQ scores, but the slight improvement in raw scores over the beef group in two subcomponents suggests that provision of fatty fish may offer benefits to certain specific aspects of cognitive function. In the FINS-KIDS trial, a similarly designed study that used the same measurements of cognitive function as the aforementioned study (the WPPSI-III and the 9-HPT), provision of herring and mackerel to 4–6-year-old children in Norway increased total raw scores on the WPPSI-III when analyses were adjusted for dietary compliance [57]. When looking at individual sub-tests of the WPPSI-III, the fatty fish group showed improved performance on three of the sub-tests: the symbol search test, which is one of the sub-tests that showed improved scores in the Atlantic salmon trial, and the vocabulary and block design sub-tests. The FINS-KIDS trial also examined whether there was any association between changes in total hair mercury concentrations after the fatty fish intervention and cognition [60] and found that while the herring/mackerel intervention did increase mercury levels, the values remained below a level of concern and were not associated with cognitive function.

#### 3.7.4. Effects of Nutritional Intervention Combined with Psychosocial Stimulation

One RCT [63] measured the impact of dietary intervention combined with psychosocial stimulation on children’s cognitive development. This trial was conducted in Indonesia and included children 3–5 years of age who had a below-average level of stimulation in the home. Children receiving a fortified milk powder supplement who performed psychosocial stimulation activities three times per week [63] demonstrated a larger increase in the full-scale IQ composite score component of the WPPSI-IV compared to the control group; however, none of the IQ sub scores were different between the intervention and control group. Parents of children in the experimental group additionally reported a larger reduction in attention problems compared to that reported for children in the control group.

## 4. Discussion

This review suggests nutritional interventions significantly improve cognitive outcomes of undernourished preschool-age children. Trials conducted in LMICs demonstrated that nutrient-deprived children who received dietary interventions consistently showed improvements in cognition. However, caution should be taken when interpreting findings due to the clinical heterogeneity of eligible studies. Metallinos-Katsaras et al. [58] suggests preschool children with anemia who receive iron supplementation can process information faster while making fewer mistakes. These findings are consistent with another systematic review [69]. According to Roberts et al. [66], nutrient-deprived children in the intervention group of the NEWSUP study showed improved working memory. Ogunlade et al. [64] stated that undernourished children who consumed point-of-use fortification displayed superior information analysis and problem-solving skills and improved mental processing abilities. Black et al. [65] noted that children attending low-quality preschools who received a multiple micronutrient powder with their meals improved expressive language. Considering the detrimental effects of nutrient deficiencies on children’s cognitive, social, and emotional skills, multiple micronutrient supplementation is a promising intervention to re-establish nutrient balance and to increase the developmental potential of undernourished preschool-age children. Although the first 1000 days of life are well-established as the most critical period for brain development and growth in a child’s life, this review suggests the second 1000 days to also be a critical period in cognitive development. Thus, preschool-age children at risk for nutrient deficiencies who receive dietary intervention can further develop their cognitive abilities. Although promising outcomes can be achieved from nutrient supplementation interventions, operational, monetary, and sustainability challenges might hinder supplement-based trial design and implementation. Additionally, multiple-micronutrient fortification intervention trials discussed in this review yielded positive effects on cognition in the short-term, and little research exists regarding the long-term efficacy of such interventions.

This review indicates well-nourished children benefit from increased fish consumption. Demmelmair et al. [56] found that healthy children in the intervention group who consumed fish increased their full-scale IQ scores from baseline, and Øyen et al. [57] found that children displayed improved processing speed abilities with fish consumption. The benefits of increased fish consumption on the cognitive development of preschool-age children identified in this review are consistent with the findings of another review [70]. On the other hand, three studies that implemented single nutrient supplement-based interventions, B vitamin sachets [55], iodized salt [61], and guava [62] showed no significant benefits associated with supplementation on cognitive outcomes of preschool-aged children. Of the studies included in this review, only 25% measured the effect of real food interventions on children’s cognition, and of these, all were conducted in high-income countries and solely interested in the impact of fatty fish and DHA on children’s cognition.

To our knowledge, there were no trials designed to explore the benefits of increased vegetable and fruit consumption on the cognitive performance of preschool-age children. Furthermore, no study measured the effect of improved dietary diversity on preschoolers’ cognitive development. Dietary diversity is adequate when a child’s diet contains five or more of the eight recommended food groups, including breast milk, grains, roots and tubers, legumes and nuts, dairy products (milk, yogurt, cheese), flesh foods (meat, fish, poultry, liver, or other organs), eggs, vitamin A-rich fruits and vegetables, and other fruits and vegetables [53,71]. Dietary diversity is the preferred approach to improving the nutrition of a population, as it is the most sustainable and desirable approach [71]. Therefore, future trials must include interventions intended to explore the effects of increased dietary diversity on the cognitive development of young children.

Additionally, further evidence is needed to determine whether nutritional interventions combined with psychosocial stimulation result in superior cognitive outcomes. For example, Schneider et al. [63] suggested that nutritional supplementation with low-level cognitive stimulation yields improved cognitive functioning in preschool-age children. However, considering as many as 200 million children in the world lack access to sufficient nutrition and adequate cognitive stimulation [5], trials that combine psychosocial stimulations and nutrient interventions on the cognitive development of preschool-age children are needed to provide more definite evidence on the impact of a combined intervention approach on cognitive outcomes.

Further research is also required to investigate the long-term effects of multiple micronutrient supplementation in cognitive outcomes of nutrient-deprived children, as short-term benefits have been consistently demonstrated. In addition, more research is needed to explore the effects of nutritional interventions combined with psychosocial stimulation on cognitive outcomes of preschool-aged children. This evidence is essential, because cognitive development reached in preschool years often determines school readiness and predicts later life achievements [5].

## 5. Conclusions

In conclusion, the findings of this review show that nutritional interventions have a positive effect on the cognitive development of undernourished preschool-age children. Nutrient-deficient children who receive micronutrient supplementation consistently display significant advances in cognitive outcomes. Furthermore, nourished children who increase fish consumption display improvements in cognitive abilities. This review highlights the importance of adequate nutrient intake during the second 1000 days of a child’s life, and the crucial role sufficient nutrition plays in cognitive development.

## Figures and Tables

**Figure 1 nutrients-14-00532-f001:**
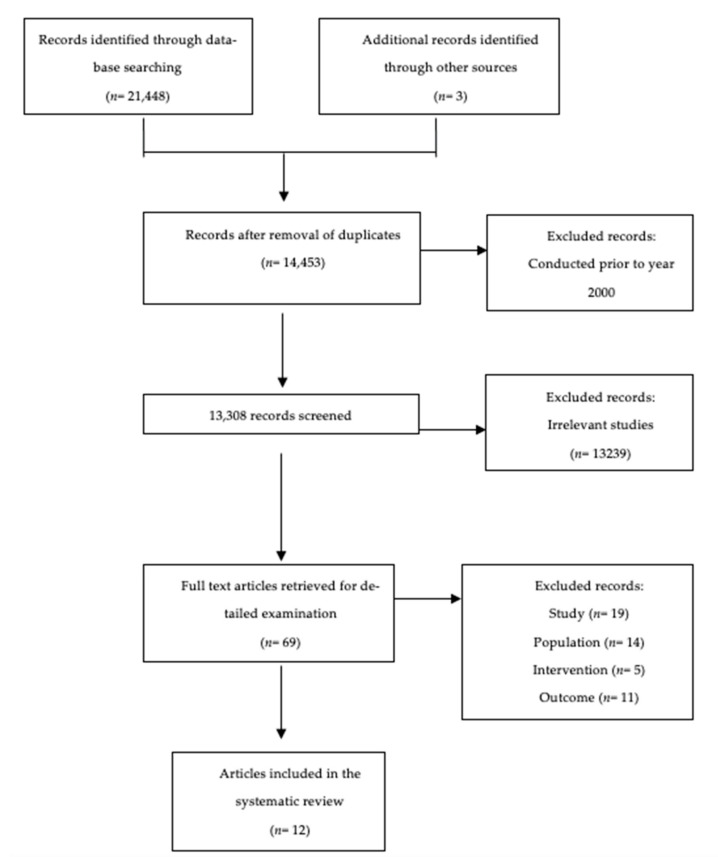
Study Selection Flow Diagram.

**Table 1 nutrients-14-00532-t001:** Inclusion and Exclusion Criteria.

Criteria	Study Design	Population	Intervention	Outcome
Include	RCTs conducted after the year 2000	Preschool Children (2–6 years of age)	Nutritional intervention including food-based, single, and multiple micronutrient supplementation intervention/s	Cognitive outcomes measured using cognitive assessment tests
		Healthy children and children suffering from undernutrition, anemia, parasitic infections, or HIV	Nutritional intervention/s provided after the first 1000days and in children <6 years of age	Cognition was measured after the first 1000 days or in children <6 years of age
	All other study designs and animal studies RCTs with a sample size <60 subjects	Newborns, infants, primary school-aged children, adolescents, adults, elderly	Nutritional intervention/s not provided to preschool-aged children	Cognitive outcomes not measured in preschool-aged children
Exclude		Children with specific diseases, such as cystic fibrosis, attention deficit hyperactivity disorder (ADHD), epilepsy, phenylketonuria, autism, and gluten-related neurological disorders		

**Table 2 nutrients-14-00532-t002:** Overview of the twelve RCTs exploring the effects of nutritional interventions on the cognitive development of preschool age children.

Reference, Year, Country	Sample Size	Age	Subject Characteristics at Baseline	Intervention Group/s	Control Group/s	Duration	Cognitive Tests	Cognitive Domain Assessed	Major Cognitive Outcomes
Rauh-Pfeiffer et al. [55], 2014, Germany	250	4–6 y	Urban area, high socioeconomic status, healthy children with low but not insuficient total folate catabolite concentrations (<34 nmol/mmol creatinine)	Children received flavorless powder containing folic acid (220 μg), riboflavin (1.1 mg), pyridoxine (0.73 mg), cobalamin (1.2 μg) and calcium lactate pentahydrate (130 mg)	Children received flavorless powder in sachets matching the intervention product in taste and appearance containing only 130 mg of calcium	3 months	WPPSI-III & (K-ABC)	Verbal I.Q., short-term memory, and processing speed	No significant difference between groups
Aboud et al. [61], 2017, Ethiopia	1602	4–6 y	Rural, low socioeconomic status, two control and four intervention districts had high UIC levels at baseline	Children had access to iodized salt for 8 to 10 months. Children received iodized salt via assistance from regular salt distributors	Children had access to non-iodized salt for 4 months, and 4 to 6 months of iodized salt. Iodized salt was introduced by market forces	10 months	WPPSI	Verbal and nonverbal reasoning, and school readiness	No significant difference between groups
Demmelmair et al. [56], 2019, Germany	205	4–6 y	Urban area, high socioeconomic status, healthy children	Children received three meals weekly containing 50 g Atlantic salmon per meal	Children received three meals weekly containing 50 g of meat per meal	4 months	WPPSI-III & 9-HPT	Fine-motor skills, verbal reasoning, vocabulary, word and matrix reasoning, picture concepts, processing speed, coding, and symbol search	Intervention children displayed superior outcomes in WPPSI-III FIQ and PIQ. No significant changes were found in the WPPSI-III I.Q. scale scores between groups
Choudhury et al. [62], 2021, India	352	3–5 y	Rural, low socioeconomic status, ICDS beneficiaries only, and children with hemoglobin concentration >7 g/dL	Children received 25 g of guava with a supplementary meal (guava group) or 25 g of banana with a meal (banana group)	Children did not receive any fruits with the meal (cucumber was given with meal if caregivers of participants wished)	8 months	MSEL	Visual reception, expressive language development, and fine-motor coordination	No significant difference between groups
Øyen et al. [57], 2018, Norway	232	4–6 y	Urban area, high socioeconomic status, healthy children	Children received three lunch meals per week with fatty fish (herring/mackerel), with a mean (SD) of 15.2 (14.2) mg/g EPA + DHA	Children received three lunch meals per week with meat (chicken/lamb/beef) with mean (SD) of 0.21 (0.15) mg/g EPA + DHA	4 months	WPPSI-III & 9-HPT	Fine-motor skills, verbal reasoning, vocabulary, word and matrix reasoning, picture concepts, processing speed, coding, and symbol search	Intervention children improved speed of processing and fine-motor coordination in a sub-analysis adjusting for dietary compliance. No significant difference was found between in main analysis of total I.Q. scores (WPPSI-III)
Schneider et al. [63], 2018, Indonesia	192	3–5 y	Urban, an upper middle-income country, children with a below-average level of stimulation at home, normal cognitive development, and weight for height within 2 SD from the median z-score	Children consumed milk powder (477.7kcal) fortified with zinc (8), iron (11.4), magnesium (141), thiamin (1), niacin (11), pyridoxine (1.7), biotin (0.0177), Vitamin C (97.3), AHA (556.6) mg/100 g, and performed psychosocial stimulation 3 times a week	Children consumed 72 g of unfortified skimmed milk powder diluted in 180 mL of warm water (467.8 kcal) and did not receive psychosocial stimulation	6 months	WPPSI-IV, CBCL 1.5–5 & PICCOLO	Cognitive functioning, cognitive development, memory, language, psychomotor skills, problem-solving, and attention	Children in the intervention group displayed increased cognitive performance and full-scale I.Q. composite score (WPPSI-IV), and reduction in attention problems (CBCL 1.5–5)
Metallinos-Katsaras et al. [58], 2004, Greece	124	3–4 y	Urban, high-income country, children with birth weight ≥ 2500 g, I.Q. ≥ 1 s.d. below the age-adjusted mean, blood Pb ≤ 200 ppb, weight and head circumference for the age ≥ 10th percentile	Children received 15 mg of iron and a multivitamins supplement (MV) five days per week at their respective day care center	Children received only the multivitamins supplement (MV) five days per week at their respective day care center	2 months	Simple reaction time test, CPT & O.L. tasks	Speed of information processing, speed of discrimination, the accuracy of discrimination, and rate of conceptual learning	Iron-deficient children who received iron supplementation showed 14% increase in discrimination speed and 8% improvement in the accuracy domain. No effects in cognitive functioning were seen in good iron status children
Ogunlade et al. [64], 2011, South Africa	151	3–6.5 y	Urban, low socioeconomic status, children with Hb ≤ 12.5 g/dl, all children received anthelmintic	Children consumed 35 g of stiff maize-meal porridge with added micronutrient powder (8 g) containing amylase-rich light malted barley flour 5 days per week	Children consumed 28 g of soft maize-meal with added placebo powder (8 g) 5 days per week	2.7 months	MPI, KABC-II, Atlantis & NVI	Learning abilities, sequential and simultaneous processing, and intellectual functioning	Intervention children showed significantly higher conceptual thinking abilities, higher MPI and NVI scores
Ryan and Nelson [59], 2008, USA	175	4 y	High-income country, healthy children consuming <6 oz of fish per week, English speakers, between 10th and 95th percentiles for weight and height, and currently not taking LC-PUFA supplements or consuming LC-PUFA fortified foods	Children received 400 mg of DHA supplementation as two 200-mg bubblegum-flavored softgel chewable	Children received capsules or placebo of high-oleic sunflower oil supplied as 2 soft capsules	4 months	PPVT & kCPT	Memory, attention, vocabulary, processing speed, response time, listening skills, and verbal ability	There was no significant difference between groups in Leiter-R Test of Sustained Attention, (PPVT), Day-Night Stroop Test, and (kCPT). Regression analysis showed a significant positive association between levels of DHA in capillary whole blood and improved listening comprehension and vocabulary (PPVT)
Black et al. [65], 2021, India	321	3–5 y	Rural, low socioeconomic status, children living in a district with prevalence of anemia >70%, >50% of children consumed <50% of the recommended intake of several essential micronutrients, high-quality and low-quality preschools were included	Children received 300g of cooked food fortified with MNP (13 mg iron, 5 mg zinc, 20 μg folic acid, 150 μg vitamin A, 20 mg vitamin C, 0.5 μg vitamin B-12, and 0.5 mg riboflavin	Children received 300 g of cooked food containing 0.5 mg riboflavin (no effects on outcome measures)	8 months	MSEL & BSID-III	Fine motor skills, gross motor skills, visual reception, receptive language, expressive language, and social-emotional behaviors	For children attending low-quality preschools, MNP fortification improved expressive language and marginally improved inhibitory control and social-emotional development in comparison to children attending control low-quality preschools. MNP fortification did not impact any area of cognitive development in children attending high-quality preschools
Kvestad et al. [60], 2018, Norway	232	4–6 y	High-income country, healthy children with no food allergies	Children received lunch meals containing 50–80 g of fatty fish (herring/mackerel) three times per week	Children received lunch meals containing 50–80 g of meat (chicken/lamb/beef) three times per week	4 months	WPPSI-III	Information, vocabulary, block design, word and matrix reasoning, picture concepts, coding, and symbol search	No significant difference between groups
Roberts et al. [66], 2020, Guinea-Bissau	1059	1.3–7 y	Rural, low socioeconomic status, children living in one of the 10 rural villages in the Oio and Cacheu regions of Guinea-Bissau, children with severe acute malnutrition or relevant food allergies were excluded from the study	One group of children received NEWSUP (≈310 kcal) for breakfast as a raw paste containing 98% of recommended daily micronutrients for children under 4 y, the second group received FBF (≈310 kcal) for breakfast served as a corn soy blend with cooked porridge, fortified oil, sugar, and salt containing an average of 16% recommended daily micronutrients	Children received white rice cooked in water, soybean oil, and salt (≈310 kcal) containing an average of 1% of recommended daily micronutrients	5.7 months	Working Memory Task Test	Working memory	Intervention children younger than 4y receiving NEWSUP displayed increased working memory compared to control children

Randomized Controlled Trials (RCTs); Wechsler Pre-school and Primary Scale of Intelligence (WPPSI); Kaufman Assessment Battery for Children (KABC); Intelligence Quotient (I.Q.); Urine Iodine Concentration (UIC); 9-Hole Peg Test (9-HPT); Full Scale I.Q. (FIQ); Performance Intelligence Quotient (PIQ); Integrated Child Development Services (ICDS); Mullen Scales of Early Learning (MSEL); Standard Deviation (SD); Eicosapentaenoic acid (EPA); Docosahexaenoic acid (DHA); Calories (kcal); Child Behavior Checklist (CBCL); Parenting Interactions with Children: Checklist of Observations Linked to Outcomes (PICCOLO); Multivitamin (MV); Cognitive Performance Test (CPT); Oddity Learning (O.L.); Multidimensional Prognostic; Index (MPI); Nonverbal Index (NVI); Long-chain polyunsaturated fatty acids (LC-PUFAs); Polyunsaturated Fatty Acids (PUFAs); Peabody Picture Vocabulary Test (PPVT); Conners Kiddie; Continuous Performance Test (kCPT); Bayley Scales of Infant Development (BSID); Multiple-micronutrient (MMN); Point-of-use multiple micronutrient powder (MNP). y: years old.

**Table 3 nutrients-14-00532-t003:** Quality Criteria Checklist (QCC; risk of bias) assessment of each study included in the review [55,56,57,58,59,60,61,62,63,64,65,66].

First Author, Year of Publication (Reference)	Rauh-Pfeiffer, 2014 [55]	Demmelmair, 2019 [56]	Ogunlade, 2011 [64]	Choudhury, 2021 [62]	Black, 2021 [65]	Kvestad, 2018 [60]	Roberts, 2020 [66]	Katsaras, 2004 [58]	Aboud, 2017 [61]	Øyen, 2018 [57]	Schneider, 2018 [63]	Ryan, 2008 [59]
Primary Research QCC												
1. Was the research question clearly stated?	Y	Y	Y	Y	Y	Y	Y	Y	Y	Y	Y	Y
2. Was the selection of study subjects/patients free from bias?	Y	Y	Y	Y	Y	Y	Y	Y	Y	Y	Y	Y
3. Were study groups comparable?	Y	Y	N	Y	Y	Y	Y	Y	N	Y	Y	Y
4. Was method of handling withdrawals described?	Y	Y	Y	N	Y	Y	Y	N	N	N	N	Y
5. Was blinding used to prevent introduction of bias?	Y	N	Y	Y	Y	Y	N	Y	N	Y	N	Y
6. Were intervention/exposure factor or procedure and any comparison(s) described in detail?	Y	N	Y	Y	Y	Y	Y	Y	Y	Y	Y	Y
7. Were outcomes clearly defined and the measurements valid and reliable?	Y	Y	Y	Y	Y	Y	Y	Y	Y	Y	Y	N
8. Was the statistical analysis appropriate for the study design and type of outcome indicators?	Y	Y	Y	Y	Y	N	Y	Y	Y	Y	Y	Y
9. Were conclusions supported by results with biases and limitations taken into consideration?	N	Y	Y	Y	Y	Y	Y	Y	Y	Y	Y	Y
10. Is bias due to study’s funding or sponsorship unlikely?	Y	Y	Y	Y	Y	Y	Y	?	Y	Y	N	Y
OVERALL QUALITY	(+)	(+)	(Ø)	(Ø)	(+)	(+)	(+)	(Ø)	(Ø)	(Ø)	(Ø)	(+)

Plus/positive (+); neutral (Ø).

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
