# Peer review of "The Effects of Nutritional Interventions on the Cognitive Development of Preschool-Age Children: A Systematic Review"

_nutrients, 2022, doi:10.3390/nu14030532_

Round 1

Reviewer 1 Report

Background/Methods:

Roberts et al. present a well-written systematic review that covers in important topic, i.e., can nutritional interventions improve the cognitive development of preschool-aged children, and focusing on the second 1000 days. The topic is lacking in good data, and it is good to know that 12 trials since 2000 were found.  PRISMA guidelines & PICO framework followed for studies between 2000-2021. Although in this framework would be more contemporary and potentially better designed, not clear why this was chosen. QCC for primary research used. Forward and backward searches of reference lists were included. Team worked closely and conflicts were identified. Rationale for exclusion included RCT <60 subjects, again not clear why this was selected and should be noted in methods. A little more discussion about heterogeneity may be in order, because these are heterogeneous studies.

Rationale: Unclear writing of why trials targeting to the first 1000 days were excluded or trials that provided parental education as the sole intervention. These rationale should be emphasized a bit better and expanded upon in the Discussion.

Discussion:

The conclusions for micronutrient and fish supplementation may be more robust than others due to sample size with these and more numbers. It would be helpful to discuss that in more detail. This should be emphasized in the Discussion.

One major limitation is the diversity of the studies in both population and nutrients, and this should be emphasized a bit more.

The Discussion should also include the 1000 days. Although the focus is of the second 1000 days, it seems that the study should summarize findings for comparison purposes in the first 1000 days, when recent meta-analyses have covered this well. This would place the second 1000 days in better context. Note that different brain processes are seen in these two blocks. For example, recent meta-analyses in Nutrition examined this and are worth siting. For example, here are just 2, but there are more.

Choline, Neurological Development and Brain Function: A Systematic Review Focusing on the First 1000 Days.  Derbyshire E, Obeid R. Nutrients. 2020 Jun 10;12(6):1731. doi: 10.3390/nu12061731. PMID: 32531929

The Effect of Low Dose Iron and Zinc Intake on Child Micronutrient Status and Development during the First 1000 Days of Life: A Systematic Review and Meta-Analysis. Petry N, Olofin I, Boy E, Donahue Angel M, Rohner F. Nutrients. 2016 Nov 30;8(12):773. doi: 10.3390/nu8120773. PMID: 27916873

Studies were not necessarily powered to evaluate the effect on high- and low-quality preschools or other efforts to improve cognitive function, but some of the studies examined that as a moderator of effect. Thus, it may be worth expanding upon this in the Discussion. For example, the study is reaching a bit to say in line 372 that the interventions can reduce the “burden of disparity among privileged and underprivileged children” without placing other environmental influences in context.

More interesting and worth emphasizing in the Discussion is the fish consumption in well-nourished children.

Minor points:

Line 298, “trial where the children erred somewhat younger [58] the cognitive benefits….”

It may be best to describe the nutritional strengths of Guava, i.e., antioxidants, vitamin C & A, lycopene, calcium, manganese, and potassium.

Author Response

Response to Reviewer 1:

My co-authors and I are pleased to have the opportunity to respond to your comments regarding our manuscript.  We appreciate the prompt reading of our manuscript your constructive and encouraging comments. This letter describes our responses in light of your comments. 

Point 1:

Roberts et al. present a well-written systematic review that covers in important topic, i.e., can nutritional interventions improve the cognitive development of preschool-aged children, and focusing on the second 1000 days. The topic is lacking in good data, and it is good to know that 12 trials since 2000 were found.  PRISMA guidelines & PICO framework followed for studies between 2000-2021. Although in this framework would be more contemporary and potentially better designed, not clear why this was chosen. QCC for primary research used. Forward and backward searches of reference lists were included. Team worked closely and conflicts were identified. Rationale for exclusion included RCT <60 subjects, again not clear why this was selected and should be noted in methods. A little more discussion about heterogeneity may be in order, because these are heterogeneous studies.

Response 1:  The Quality Criteria Checklist (QCC) for Primary Research from the American Dietetic Association has been increasingly selected as an effective tool to assess study quality.

For example, here are recent systematic reviews that adopted the tool:

Taylor RM, Fealy SM, Bisquera A, Smith R, Collins CE, Evans TJ, Hure AJ. Effects of Nutritional Interventions during Pregnancy on Infant and Child Cognitive Outcomes: A Systematic Review and Meta-Analysis. Nutrients. 2017 Nov 20;9(11):1265. doi: 10.3390/nu9111265. PMID: 29156647; PMCID: PMC5707737.

Abbate M, Gallardo-Alfaro L, Bibiloni MDM, Tur JA. Efficacy of dietary intervention or in combination with exercise on primary prevention of cardiovascular disease: A systematic review. Nutr Metab Cardiovasc Dis. 2020 Jun 25;30(7):1080-1093. doi: 10.1016/j.numecd.2020.02.020. Epub 2020 Mar 17. PMID: 32448717.

Deliens T, Van Crombruggen R, Verbruggen S, De Bourdeaudhuij I, Deforche B, Clarys P. Dietary interventions among university students: A systematic review. Appetite. 2016 Oct 1;105:14-26. doi: 10.1016/j.appet.2016.05.003. Epub 2016 May 13. PMID: 27181201.

Dalwood P, Marshall S, Burrows TL, McIntosh A, Collins CE. Diet quality indices and their associations with health-related outcomes in children and adolescents: an updated systematic review. Nutr J. 2020 Oct 24;19(1):118. doi: 10.1186/s12937-020-00632-x. PMID: 33099309; PMCID: PMC7585689.

Additional discussion about the heterogeneity of studies was added under results section (204-207) and discussion (384-385).

Point 2:

Rationale: Unclear writing of why trials targeting to the first 1000 days were excluded or trials that provided parental education as the sole intervention.

Response 2: The introduction section of the review was further expanded to explain this.

Point 3:

Discussion: The conclusions for micronutrient and fish supplementation may be more robust than others due to sample size with these and more numbers. It would be helpful to discuss that in more detail. This should be emphasized in the Discussion.

Response 3: Suggestion acknowledged. Further emphasis was not included in discussion due to the short duration of trials.

Point 4:

One major limitation is the diversity of the studies in both population and nutrients, and this should be emphasized a bit more.

Response 4: Further discussion of study heterogeneity and interpretation challenges was added to the results and discussion sections of the review.

Point 5:

The Discussion should also include the 1000 days. Although the focus is of the second 1000 days, it seems that the study should summarize findings for comparison purposes in the first 1000 days, when recent meta-analyses have covered this well. This would place the second 1000 days in better context. Note that different brain processes are seen in these two blocks. For example, recent meta-analyses in Nutrition examined this and are worth siting. For example, here are just 2, but there are more.

Choline, Neurological Development and Brain Function: A Systematic Review Focusing on the First 1000 Days.  Derbyshire E, Obeid R. Nutrients. 2020 Jun 10;12(6):1731. doi: 10.3390/nu12061731. PMID: 32531929

The Effect of Low Dose Iron and Zinc Intake on Child Micronutrient Status and Development during the First 1000 Days of Life: A Systematic Review and Meta-Analysis. Petry N, Olofin I, Boy E, Donahue Angel M, Rohner F. Nutrients. 2016 Nov 30;8(12):773. doi: 10.3390/nu8120773. PMID: 27916873

Response 5: More details were added to the discussion to summarize findings in the first 1000 days for comparison  

Point 6:

Studies were not necessarily powered to evaluate the effect on high- and low-quality preschools or other efforts to improve cognitive function, but some of the studies examined that as a moderator of effect. Thus, it may be worth expanding upon this in the Discussion. For example, the study is reaching a bit to say in line 372 that the interventions can reduce the “burden of disparity among privileged and underprivileged children” without placing other environmental influences in context.

Response 6: Thank you for this feedback. Changes were made to the discussion and conclusion in response to this point.

Point 7:

Line 298, “trial where the children erred somewhat younger [58] the cognitive benefits….”

Response 7: Thank you, this was corrected.

Point 8:

It may be best to describe the nutritional strengths of Guava, i.e., antioxidants, vitamin C & A, lycopene, calcium, manganese, and potassium.

Response 8: We agree; we added details about the nutritional value of guava (see lines 301-306).

Reviewer 2 Report

Thank you for the opportunity to review this important and well-written review. I very much enjoyed reading this review and consider it an important contribution to the field of nutrition and child development which is, indeed, very focused on the first 1000 days.

Introduction

Well-written and relevant.

Search strategy, inclusion criteria, and data extraction

Overall well-executed and detailed search.

Inclusion criteria

What was the motivation behind only including studies after 2000? Why would older studies not be deemed relevant? This seems to have excluded some important studies like the combined nutrition and stimulation studies from Jamaica in the 1990s.

The age restriction refers to the assessment of the outcome, not the timing of the intervention. I think the breakdown regarding the timing of the intervention might be more important, but I do appreciate this approach focusing on the timing of the outcome as being relevant as well. If nothing else, a breakdown in the discussion of whether the timing of the intervention mattered would be enlightening.

Data extraction

Did the two reviewers of the full-text records review all records or an overlapping subsample? Did you quantify their inter-rater agreement?

Results

It would be helpful if the outline of findings stated where the study was conducted. I know that this information is in the table, but by including such information in e.g. parentheses the reader won’t have to refer to the table while reading.

Findings:

The review covers a complex set of studies that vary in many different ways and the review of findings can therefore be meaningfully broken down in different ways. Some important groupings of studies include; timing of the intervention (See comment regarding inclusion criteria), type of intervention (current focus), child nutrition status before the intervention, context (LMIC vs. HIC), combined versus stand-alone nutrition intervention. I understand that it would be repetitive to discuss all of these breakdowns in the discussion, but I wonder if it would be possible to create a table or graphic that illustrates finds considering these groupings/ breakdowns?

I am surprised by the lack of combined nutrition and psychosocial stimulation studies. Is this because of the age criteria or time cut off? I have copied in some links of the study from Jamaica and the PEDS trial from Pakistan both of which included both. Even if these studies don’t meet the inclusion criteria here, the results from older studies or studies with slightly different criteria could be mentioned in the discussion.

Links:

Jamaica study (excluded due to publication date I assume?)

https://pubmed.ncbi.nlm.nih.gov/1676083/

PEDS Study

https://www.sciencedirect.com/science/article/pii/S2214109X16301000

Author Response

Response to Reviewer 2:

My co-authors and I are pleased to have the opportunity to respond to your comments regarding our manuscript.  We appreciate the prompt reading of our manuscript your constructive and encouraging comments. This letter describes our responses in light of your comments. 

Point 1:

Thank you for the opportunity to review this important and well-written review. I very much enjoyed reading this review and consider it an important contribution to the field of nutrition and child development which is, indeed, very focused on the first 1000 days.

Introduction Well-written and relevant.

Search strategy, inclusion criteria, and data extraction Overall well-executed and detailed search.

Response 1: Thank you for this feedback.

Point 2:

Inclusion criteria

What was the motivation behind only including studies after 2000? Why would older studies not be deemed relevant? This seems to have excluded some important studies like the combined nutrition and stimulation studies from Jamaica in the 1990s.

Response 2: The motivation behind including studies after year 2000 was to evaluate the nutritional interventions implemented in the past 20 years targeted to improve cognitive outcomes of preschool-age children.

Point 3:

The age restriction refers to the assessment of the outcome, not the timing of the intervention. I think the breakdown regarding the timing of the intervention might be more important, but I do appreciate this approach focusing on the timing of the outcome as being relevant as well. If nothing else, a breakdown in the discussion of whether the timing of the intervention mattered would be enlightening. 

Response 3: Thank you for your comment. Suggestion acknowledged but kindly declined at this time

Point 4:

Data extraction

Did the two reviewers of the full-text records review all records or an overlapping subsample? Did you quantify their inter-rater agreement?

Response 4: The two reviewers reviewed all records, however inter-rater agreement was not quantified.

Point 5:

Results

It would be helpful if the outline of findings stated where the study was conducted. I know that this information is in the table, but by including such information in e.g. parentheses the reader won’t have to refer to the table while reading.

Response 5: Thank you for your comment. Suggestion acknowledged.

Point 6:

Findings:

The review covers a complex set of studies that vary in many different ways and the review of findings can therefore be meaningfully broken down in different ways. Some important groupings of studies include; timing of the intervention (See comment regarding inclusion criteria), type of intervention (current focus), child nutrition status before the intervention, context (LMIC vs. HIC), combined versus stand-alone nutrition intervention. I understand that it would be repetitive to discuss all of these breakdowns in the discussion, but I wonder if it would be possible to create a table or graphic that illustrates finds considering these groupings/ breakdowns?

Response 6: Thank you for the suggestion. Although this recommendation would add substantial value to this review, it will not be possible due to the time constraint of the revision period.

Point 7:

I am surprised by the lack of combined nutrition and psychosocial stimulation studies. Is this because of the age criteria or time cut off? I have copied in some links of the study from Jamaica and the PEDS trial from Pakistan both of which included both. Even if these studies don’t meet the inclusion criteria here, the results from older studies or studies with slightly different criteria could be mentioned in the discussion.

Links:

Jamaica study (excluded due to publication date I assume?)

https://pubmed.ncbi.nlm.nih.gov/1676083/

PEDS Study

https://www.sciencedirect.com/science/article/pii/S2214109X16301000

Response 7: Yes, the limited number of these studies is directly related to the age criteria (preschool-age). This review intended to shine light to the importance of adequate nutrition during the preschool-age years in addition to the first 1000 days of life. Thank you for taking the time to include the links to relevant studies.
